# Carbapenem-Resistant *Enterobacteriaceae* (CRE) among Children with Cancer: Predictors of Mortality and Treatment Outcome

**DOI:** 10.3390/antibiotics12020405

**Published:** 2023-02-17

**Authors:** Youssef Madney, Shaimaa Aboubakr, Reham Khedr, Hanafy Hafez, Naglaa Ahmed, Khaled Elsheshtawy, Mervat Elanany, Abdelhamid Salahelden, Lobna Shalaby, Ola Galal Behairy

**Affiliations:** 1Department of Pediatric Oncology, National Cancer Institute, Cairo University, Cairo 12613, Egypt; 2Department of Pediatric Oncology, Children’s Cancer Hospital Egypt, Cairo 57357, Egypt; 3Department of Paediatrics, Faculty Of Medicine, Benha University, Benha 15881, Egypt; 4Department of Clinical Pharmacy, Children’s Cancer Hospital Egypt, Cairo 57357, Egypt; 5Department of Clinical Research, Children’s Cancer Hospital Egypt, Cairo 57357, Egypt; 6Department of Clinical Microbiology, Faculty Of Medicine, Cairo University and Children’s Cancer Hospital Egypt, Cairo 57357, Egypt

**Keywords:** carbapenem-resistant Enterobacteriaceae (CRE), children, cancer, predictors of mortality

## Abstract

Carbapenem-resistant Enterobacteriaceae (CRE) is an important emerging threat among pediatric cancer patients, with a high mortality rate. This retrospective study included all pediatric cancer patients with (CRE) bloodstream infections (BSIs) at a children’s cancer hospital in Egypt (2013–2017). Two hundred and fifty-four pediatric cancer patients with CRE BSI were identified; 74% had hematological malignancies, and 26% had solid tumors. Acute myeloid leukemia was the most common hematological malignancy (50%). The main clinical features for acquiring CRE-BSI were previous antibiotics exposure (90%), profound neutropenia (84%), prolonged steroid use (45%), previous colonization with a resistant pathogen (35%), ICU admission within 90 days (28%), and central venous catheter use (24%). *E. coli* was the most common isolated pathogen (56%), followed by *Klebsiella pneumoniae* (37%). All isolates were resistant to carbapenem with an MIC < 4–8 μg/mL in 100 (45%) and >8 μg/mL in 153 (55%). The overall mortality rate was 57%, and 30 day mortality was reported in 30%. Upon multivariate analysis, for the patients with *Klebsiella pneumoniae* BSI, carbapenem resistance with an MIC > 8 μg/mL and associated typhlitis or pneumonia were predictors of poor outcome. In conclusion, CRE-BSI is a major threat among pediatric cancer patients in limited resource countries with limited options for treatment. Antimicrobial stewardship for early detection through routine screening, adequate empirical treatment, and timely adequate therapy may impact the outcome for such high-risk patient groups.

## 1. Introduction

Carbapenem-resistant Enterobacteriaceae (CRE) infections increase in children and are associated with poor clinical outcomes [1,2]. The most common CRE in children include *Escherichia coli (E. coli*), *Klebsiella pneumoniae*, and *Enterobacter* species [1]. The Centers for Disease Control and Prevention define CRE as any Enterobacteriaceae exhibiting carbapenem resistance with a meropenem, imipenem, or doripenem minimum inhibitory concentration (MIC) ≥ 4 μg/mL or an ertapenem MIC ≥ 2 μg/mL [3]. The main resistance mechanism in CRE is carbapenemases production, mainly K. pneumoniae carbapenemase (KPC), New Delhi metallo-β-lactamases (NDM), Verona integron-encoded metallo-β-lactamases (VIM), and oxacillinase (OXA)-48-like enzymes (OXA-48), which display different geographical variations [3]. There is a difference in carbapenemase distribution; KPC is the main carbapenemase in North America and Europe, while it is uncommon in the Middle East, where MBLs are reported to be more common [4]. CRE infections in patients with hematological malignancies undergoing intensive chemotherapy are very challenging [5,6].

An usnderlying malignancy, intensive chemotherapy with subsequent neutropenia, prolonged hospitalization, and gastrointestinal mucositis are risk factors for CRE bacteremia [7,8]. Given the lack of pediatric studies, treating CRE infections in children is complex and based mainly on adult data [3]. Treatment failures, inappropriate initial empirical treatment, and delayed targeted therapy for CRE infections are common in febrile neutropenic cancer patients [9,10]. The mortality rate associated with CRE bacteremia in neutropenic hematological cancer patients is approximately 60% [7,8]. 

Data on survival outcomes and prognostic factors for CRE bacteremia in children with cancer in areas with limited resources are still lacking. The pattern of CRE infections in children in regions with limited resources differs in terms of prevalence, outcome, available therapeutics, and management strategies. Recent guidelines recommend using new drugs not available in countries with limited diagnostics and treatment resources. This study describes the clinical features, microbiological sensitivity, and outcome of CRE bloodstream infections among children with cancer to identify mortality predictors among such high-risk groups.

## 2. Results

### 2.1. Clinical Characteristics of the Patients 

The study included 254 children with cancer with a CRE bloodstream infection (BSI) from 2013 to 2017. During the study period, 9600 children were diagnosed with malignancy and treated in CCHE. One hundred and eight thousand four hundred and sixty-two (108,462) blood cultures were conducted; 7211 (7%) showed bacterial growth, and 3648 (46%) of the bacterial growths were Gram-negative. The most common identified Gram-negative pathogens were *E. coli* (n = 1372), *K. pneumoniae* (n = 866), pseudomonas aeruginosa (n = 272), and Acinetobacter baumannii (n = 263) (Appendix A). The median age was six years (range: 0.2–18), and the patients were predominantly male (58%; 124/254). Most of the patients (74%; 188/254) had hematological malignancies, while 26% (66/254) had solid tumors. Acute myeloid leukemia was the most frequent hematological malignancy (54%; 101/188). Approximately half of the patients (51%; 130/254) had a refractory or relapsed disease. At the time of bacteremia, 87% (220/254) of the patients were hospitalized in the medical ward, while 13% (34/254) were in the intensive care unit (ICU). 

The main clinical features reported among patients with CRE Infection were profound neutropenia (absolute neutrophil count (ANC) less than 100 cells/microliter) with a duration of more than seven days (80%; 203/254); previous exposure to antibiotics in the last 90 days (90%; 229/254); steroid use (45%; 115/254); quinolone prophylaxis (30%; 76/254); ICU admission (28%; 71/254); and central venous catheter (24%; 60/254). Colonization with CRE was reported in 35% (90/254) of the patients. The clinical characteristics of the patients with CRE bacteremia are shown in Table 1.

### 2.2. Isolated Pathogens and Resistance Pattern

The most common isolated bacteria were *E. coli* (60%; 151/254), followed by *K. pneumoniae* (37%; 94/254) and *Enterobacter cloacae* (3%; 9/254). A simultaneous BSI and a documented site of infection were detected in 56% (180/254) of the patients. The most common infection sites were pneumonia (27%; 68/254), skin and soft tissue (20%; 50/254), typhlitis (18%; 47/254), and urinary tract (5%; 15/254). Carbapenem resistance was reported in all patients with an MIC less than 8 μg/mL (45%) and more than 8 μg/mL (55%). Aminoglycoside resistance was reported in 46.5%, and 87% were resistant to quinolones. Colistin resistance was reported in 1% (two isolates), while no resistance was reported to tigecycline. Thirty-two patients (12%) had recurrent CRE-BSI bacteremia with the same organism and subsequent neutropenia within 90 days from the first episode.

### 2.3. Treatment of Carbapenem-Resistant Enterobacteriaceae

One hundred and forty patients (55%) received inappropriate empirical antibiotic treatment, while the remaining 114 (45%) received adequate initial treatment. The time to start active antibiotic treatment varied among the study participants. More than half (58%) started active treatment within 48 h, while the remaining (42%) started after 48 h. Seventy-three patients (29%) received active monotherapy antibiotic treatment, while 156 (61%) received combined antibiotic treatment. Among the 254 patients with CRE, 96 (38%) had septic shock and needed ICU admission and inotropic support. The overall mortality was 55%, while the 30 day mortality from BSI onset was 30% (76/254).

### 2.4. Prognostic Factors for Outcome 

Carbapenem-resistant Enterobacteriaceae (CRE) with a minimal inhibitory concentration (MIC) < 4–8 μg/mL had a better prognosis and lower 30 day mortality than CRE with an MIC > 8 μg/mL. In addition, inadequate initial antibiotic therapy was associated with higher 30 day mortality. Furthermore, it was crucial to promptly initiate an active antibiotic against CRE-BSI; mortality was higher when the active antibiotic was not started at all or delayed more than 48 h (Figure 1). Combination therapy was not associated with lower 30 day mortality than monotherapy (*p* = 0.380). A subgroup analysis according to hematological vs. solid tumors revealed the better effect of the combination therapy in patients with hematological malignancies but without statistical significance (*p* = 0.07) and no effect in patients with solid tumors (*p* = 0.240) (Appendix A). 

In univariate analysis, the patients who presented with clinical instability and needed ICU admission and inotropic support were associated with higher day 30 mortality. Regarding the type of organism, the mortality was higher with *K. pneumoniae* infections than with *E. coli* or *Enterobacter cloaca*. Moreover, patients with CRE infection with pneumonia or typhlitis were associated with higher mortality. There was no difference between the solid and hematological malignancies regarding the primary disease, but patients with refractory disease had a higher 30 day mortality (Table 2). Upon multivariate logistic regression analysis, the independent factors that significantly affected the outcomes were isolation of *K. pneumoniae* (OR: 2.2, 95% CI: 1.1–4.1, *p* = 0.020), associated typhlitis (OR: 5.1, 95% CI: 2.4–10.4, *p* < 0.001), pneumonia (OR: 2.3, 95% CI: 1.2–4.5, *p* = 0.013), and carbapenem resistance with an MIC > 8 μg/mL (OR: 2.7, 95% CI: 1.3–5.4, *p* = 0.006) (Table 3).

## 3. Discussion

We reported two hundred and fifty-four pediatric cancer patients with CRE bacteremia. Most of them (74%) had hematological malignancies. Profound, severe neutropenia and prolonged hospital stay were the main clinical features. *E. coli* was the most common pathogen (56%), followed by *K. pneumoniae* (36%). Thirty-day mortality was reported in 30%. Septic shock on presentation, inadequate empirical antimicrobial therapy, and delayed adequate active treatment more than 48 h were statistically significantly associated with poor survival.

Many studies have reported that most CRE BSI in children were hospital-acquired and associated with indwelling devices or exposure to carbapenems in the previous six months [6,7,8]. All BSI episodes in the current study were hospital-acquired (85%), while the intensive care unit represented 15%. The patients with malignancies had an increased risk of CRE bloodstream infections due to the chemotherapy-induced neutropenia, gastrointestinal mucositis, and frequent use of broad-spectrum antibacterial agents [9,10]. 

Mortality is associated with a high meropenem MIC among pediatric patients [11]. In addition, in the current study, meropenem with an MIC > 8 μg/mL was associated with a significant mortality rate. The patients with a high carbapenem MIC > 8 μg/mL in this study were associated with other poor clinical factors: *K. pneumoniae* (37%) was the most identified pathogen, associated with pneumonia (27%), and most of those patients had clinical instability with sepsis and needed ICU admission (30%). The patients with a CRE bacteremia with associated pneumonia or typhlitis had higher 30 day mortality compared to the patients with only bacteremia. This may be explained by the difficult source control in typhlitis [12] and poor tissue concentration of colistin in lung [13].

In the current study, *E. coli* was the most common isolated CRE (60%), followed by *K. pneumoniae* (37%). The patients with *K. pneumoniae* had higher 30 day mortality (48%) than those with E. coli CRE (21%). The correlation between MIC with type of organism was reported in our study and was found to be of clinical significance. The patients with *K. pneumoniae* were reported to have a higher MIC to carbapenem (80% versus. 37%). A higher *K. pneumoniae* MIC was also associated with poor prognostic factors, such as documented site infection (pneumonia and typhlitis), delayed microbiological clearance (30% versus. 16% for *E. coli)*, aminoglycoside resistance (70% versus. 30%), and greater presentation with septic shock with the need for ICU admission (42% versus. 19%).

Inadequate empirical antibiotic treatment was associated with an increased 30 day mortality [14,15]. In addition, in the current study, a higher 30 day mortality (37%) was reported in the patients with inadequate initial empirical therapy. The intestinal microbiota contains Enterobacteriaceae. Patients colonized with CRE are at an increased risk for BSI with the same organism [16,17]. Tischendorf et al. reported a 16% risk of infections among 1806 hospitalized patients colonized with CRE [18]. Acute leukemia patients are at high risk for colonization with CRE. Underlying hematological malignancies and neutropenia are associated with BSI among CRE colonized patients [14,17,19]. In the current study, 35% of the children with CRE BSI were colonized with CRE. 

Early administration of appropriate intravenous antibiotics is the cornerstone of CRE BSI initial management, while delayed, inadequate therapy is associated with lower survival [14,20]. In the current study, the patients who started active antibiotics within 48 h had a better outcome than those with delayed active treatment (24% vs. 37%, *p* = 013). These data reflect the importance of the early detection of high-risk patients for CRE BSI through screening for CRE carriers, the scoring system for high-risk CRE patients, and antibiogram-guided antimicrobial stewardship for improving outcomes. 

The optimal treatment of CRE BSI in pediatric cancer patients is still not defined [1]. Combination treatment with two or more active drugs was suggested to improve survival outcomes. Retrospective studies in adult patients demonstrated mortality benefits in patients who received combination therapy [2,21]. However, other studies indicated that the benefit is limited to patients with high mortality risk [22]. Limited data on the impact of combination therapy on survival outcomes among pediatric cancer patients with CRE are available. Although combination therapy was not more associated with lower 30 day mortality than monotherapy in this study, the subgroup analysis showed a lower mortality rate in the patients with hematological malignancies who received combination therapy.

Patients with hematological malignancies are at risk of recurrent CRE BSI due to the fact of neutropenia. However, the incidence of recurrence has not been studied extensively. Many studies among hematologic patients reported an incidence of recurrent MDR Gram-negative BSI of 15–20% [23,24]. This study reported a recurrence incidence with the same organism in thirty-two patients (12%), with subsequent neutropenia within 90 days from the first episodes. The previous history of CRE BSI should be considered in managing febrile neutropenic cancer patients.

During the study period, surveillance for CRE colonization screening was not routinely performed. However, in this study, 30% of patients with CRE BSI infections were colonized with CRE. In addition, a high mortality rate was reported among pediatric cancer patients with CRE (30%), especially if appropriate treatment was delayed. Based on the data reported in this study, colonization surveillance with a rectal swab for all leukemia patients admitted to our center with the use of colonization-guided empirical antibiotics was implemented.

The mortality rate of CRE BSI in adult hematological cancer patients was 60%, with higher rates among acute leukemia patients [10,25], while the fatality rate was 35% among children with cancer with CRE BSI [26]. This study’s mortality rate was 30%, with higher mortality mainly among leukemia patients (60%). Tumbarello et al. reported that septic shock presentation among CRE-KPC was an independent prognostic factor for mortality [2]. Moreover, in the current study, 35% of the patients had septic shock and needed ICU admission; 86% of them died within 30 days of CRE BSI onset.

A retrospective observational cohort study on the epidemiology of CRE among 310 ICUs in 72 Egyptian hospitals (2011–2017) reported that there were 1598 Enterobacteriaceae cases of which 871 (54%) were carbapenem resistant [27]. Another survey analysis evaluated the prevalence of MDR Gram-negative bacteremia in medical and surgical units in Egypt (2004–2020) and reported that carbapenem resistance was reported in 28.8–69% of Enterobacteriaceae [28]. While among Egyptian febrile neutropenic cancer patients, in a retrospective analysis of 529 patients (mean age: 16 y). BSI was detected in 195/529 (20%) patients, with 102 (52%) having Gram-negative bacteremia. Out of the 102 GNBs, 70 (68.6%) were MDR with *E. coli*, and *K.pneumoniae* was the most common (73%) detected pathogen [29]. Another study among 105 pediatric AML patients treated at the National Cancer Institute (NCI), Egypt (2014–2016), reported a mortality rate in 88 (84%) patients with MDR Gram-negative bacteremia as the main cause of death (62%). *Klebsiella pneumoniae* and *E. coli* were the most commonly identified pathogens [30].

In limited resources countries, the barriers against better the management of CRE infections were the lack of molecular diagnostics that can provide a better understanding of CRE infection resistance patterns and the lack of available new antibiotics, which can improve outcomes and decrease mortality from CRE infection in critically ill patients [27]. This study included a large number of pediatric cancer patients and offers useful insights on the prevalence, clinical features, and management of CRE-BSI in limited resource countries in which data are lacking. However, our study has some limitations. It is a retrospective study with a heterogenous patient population with different primary malignancies and chemotherapy regimens. In addition, we lacked some microbiological data, such as time to positivity of the cultures, and molecular data for detecting carbapenemases genes. 

## 4. Materials and Methods

### 4.1. Study Design 

This retrospective observational study was conducted at the Children’s Cancer Hospital Egypt (CCHE), a tertiary hospital with 200 hospitalization beds. We analyzed all CRE bloodstream infection (BSI) episodes in children (0–18 years) with cancer from 2013 to 2017. Carbapenem resistance was defined using the Clinical and Laboratory Standards Institute (CLSI) criteria. Demographic data, underlying diseases, clinical features, microbiological data with MIC, empirical and targeted antimicrobial therapy, and outcomes were collected. The primary outcome was death within 30 days after the first positive CRE blood culture.

### 4.2. Guidelines for Fever and Neutropenia Protocol at CCHE

We applied the Infectious Diseases Society of America (IDSA) guidelines for febrile neutropenia in our children with cancer. Only patients with acute myeloid leukemia (AML) received antibacterial levofloxacin and antifungal prophylaxis. From 2013 to 2015, the initial empirical antimicrobial treatment was a combination therapy with piperacillin-tazobactam and aminoglycosides. However, the shift in empirical antibiotic therapy to carbapenem/aminoglycosides since 2016 for high-risk hematological malignancies as an institutional antibiogram showed an increased incidence of ESBL Gram-negative bacteremia. 

### 4.3. Treatment Protocol for CRE

Patients with CRE-BSI were treated according to a carbapenem MIC. For an MIC ≤ 8 μg/mL, high-dose continuous infusion (40 mg/kg over 3 h) was the main active treatment, adding a second active agent (aminoglycosides or quinolones if sensitive or colistin if both were resistant). For an MIC > 8 μg/mL, colistin was the main active treatment with adding a second agent (aminoglycosides or quinolones if sensitive or tigecycline if both were resistant) [1]. The antimicrobial treatment duration for CRE-BSI was 14 days. However, for patients with a documented site of infection, such as pneumonia or GIT infection, the treatment duration differed according to clinical, microbiological, radiological, and neutrophil recovery.

### 4.4. Microbiological Detection Methods of CRE

**Blood cultures:** Pediatric blood culture bottles (BACTECTM PedsPlusTM/F, BD Diagnostics & BacT/ALERT^®^PF Plus) inoculated with patients’ blood were loaded in the corresponding blood culture system. After flagging, positive blood culture broths were sub-cultured on Columbia blood agar and MacConkey No. 2 agar plates (Oxoid™) and incubated at 37 °C. **Identification:** the isolated colonies were identified using the MALDI-TOF Vitek MS IVD system (BioMérieux; Marcy l’Etoile, France). **Antibiotic susceptibility testing:** Automated Vitek 2 Compact (BioMérieux SA, Marcy l’Étoile, France) was used on isolates on the agar plates, as recommended by the manufacturer, and interpreted according to the CLSI guidelines.

### 4.5. Definitions

**Bloodstream infection (BSI):** one or more pathogens isolated in the blood culture of a patient with symptoms and signs of infection. **Multidrug-resistant (MDR) bacteremia**: defined as nonsusceptibility to at least one agent in three or more antimicrobial categories [31]. **Carbapenem-resistant Enterobacteriaceae (CRE):** Any Enterobacteriaceae that tested resistant to at least one of the carbapenem antibiotics, with a minimum inhibitory concentration (MIC) ≥ 4 μg/mL for meropenem, imipenem, or doripenem, or an MIC ≥ 2 μg/mL for ertapenem. Treatment was considered **adequate** if at least one or more drugs had in vitro activity (MIC within the susceptible range) against the CRE isolate. Meropenem (MIC ≤ 4–8 μg/mL) was considered an effective agent based on adult studies [2,3]. **Combination therapy** was considered if the patient received two or more active antibiotics. **Sepsis or septic shock** classification was performed according to the International Consensus Conference on Pediatric Sepsis definitions [5].

### 4.6. Statistical Analysis 

Statistical analysis was performed using IBM SPSS^®^ Statistics version 22 (IBM^®^ Corp., Armonk, NY, USA). Numerical data are expressed as the means and standard deviations or medians and ranges, as appropriate. The qualitative data are expressed as frequencies and percentages. Pearson’s Chi-square test was used to examine the relationships between the qualitative variables. The quantitative data were tested for normality using the Kolmogorov–Smirnov test and Shapiro–Wilk test. For non-normally distributed quantitative data, a comparison between two groups was conducted using the Mann–Whitney test (nonparametric *t*-test). Multivariate analysis was conducted using multivariate logistic-regression for the significant factors affecting outcome (survival at 30 days) upon univariate analysis. Odds ratios (ORs) with a 95% confidence interval (CI) were used for the risk estimation. All tests were two-tailed. A *p*-value < 0.05 was considered significant.

## 5. Conclusions

In conclusion, CRE BSI is a major threat for pediatric cancer patients in countries with limited resources with limited treatment options. Antimicrobial stewardship for early detection through routine screening, adequate empirical antibiotic treatment according to local antibiogram, and combination therapy may improve the outcome for such high-risk patients.

## Figures and Tables

**Figure 1 antibiotics-12-00405-f001:**
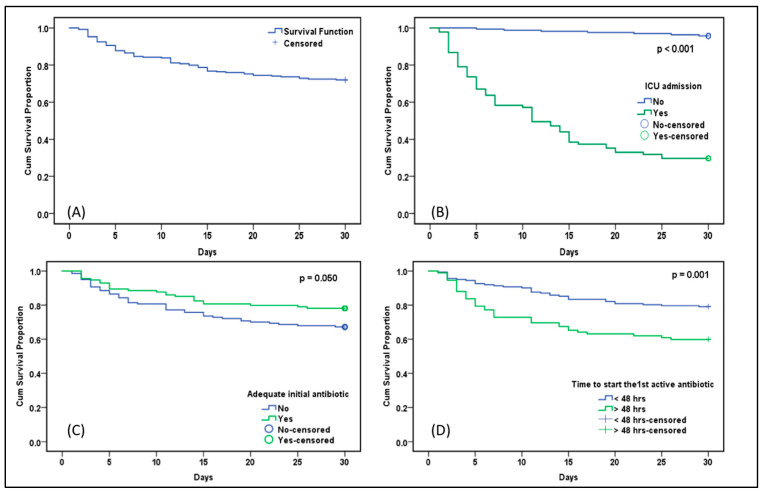
Kaplan–Meier curves of the different prognostic factors and survival at 30 days in pediatric cancer patients with CRE bacteremia: (**A**) total group patients; (**B**) need for ICU admission; (**C**) adequate initial empirical treatment; (**D**) time to start adequate treatment.

**Table 1 antibiotics-12-00405-t001:** Clinical characteristics and treatment outcome of CRE in children with cancer.

Patients with CRE-BSI	N = 254 (100%)
**Age *; median age (6 year)**	
- Infants	69(27%)
- Children	123(48.5%)
- Adolescents	62(24.5%)
**Sex**	
- Male	124(58%)
- Female	89(42%)
**Cancer type**	
**Solid tumors**	**66 (26%)**
- Neuroblastoma	29(12%)
- Sarcomas	17(7%)
**Hematological malignancies**	**188(74%)**
- AML	101(40%)
- ALL	59(23%)
- Others	28(11%)
**Diseases status**	
- Disease remission	124(49%)
- Refractory/relapsing disease	130(51%)
**Profound neutropenia** ******	213(84%)
**Previous ICU admission**	72(28%)
**Central venous line**	61(24%)
**Steroids within 30 days**	114(45%)
**Levofloxacin prophylaxis**	77(30%)
**Acquisition of BSI**	
- Intensive care unit (ICU)	34(13%)
- Medical word	220(87)
**Colonization history**	
- Not colonized	42(16.5%)
- Colonized with ESBL	62(24.4%)
- Colonized with CRE Gram-negative bacteremia	90(35.4%)
- Not conducted	60(23.6%)
**Organism type**	
- *E. coli*	151(59.5%)
- *K. pneumoniae*	94(37%)
- *Enterobacter cloacae*	9(3.5)
**Source of BSI**	
- **Bacteremia (BSI only)**	**112(44%)**
- **BSI with associated documented site infection**	**142(56%)**
- Pneumonia	68(26.8%)
- Skin and soft tissue	50(19.7%)
- Typhlitis	47(18.5)
- Urinary tract infection	15(6%)
**Time to start first active antibiotic**	
- Less than 48 h	147(58%)
- More than 48 h	107(42%)
**Adequate initial treatment**	
- Yes	114(45%)
- No	140(55%)
**Number of active targets antibiotics**	
- None	25(9.8%)
- Monotherapy	73(28.7%)
- Combination	156(61.5%)
**Septic shock and needs for ICU**	
- Septic shock	96(37.8%)
- Need for ICU admission	90(35.4%)
- Need for inotropic support	72(28.3%)
**Day 30 Mortality**	76(30%)

***** Ages categorized as per the FDA pediatric age categories. **CRE**: carbapenem-resistant Enterobacteriaceae; **BSI:** blood stream infection; **ALL**: acute lymphoblastic leukemia; **AML**: acute myeloid leukemia; **ICU**: intensive care unit. ** Absolute neutrophil count < 100/mm^3^ > **7 days**.

**Table 2 antibiotics-12-00405-t002:** Univariate analysis for predictors of mortality among pediatric cancer patients with CRE-BSI.

Children with CRE-BSI	N = 254	Alive = 178	Death = 76	OR	*p*-Value
**Refractory/relapse of primary malignancy**	130 (51%)	82 (63%)	48 (37)	2.2 (1.8–3.8)	0.005
**Type of organism**				3.2 (1.8–5.6)	<0.001
- *E. coli*	151 (59.4%)	118 (78%)	33 (21.9%)		
- *K. pneumoniae*	94 (37%)	52 (55.3%)	42 (44.7%)		
**Source**				3.5 (1.9–6.6)	<0.001
**Bacteremia only**	112 (44%)	93 (83.0%)	19 (17.0%)		
**Associated documented site of infection**	142 (60%)	85 (59.9%)	57 (40.1%)		
- Pneumonia	68 (26.7%)	35 (51.5%)	33 (48.5%)	3.3 (1.8–6.0)	<0.001
- Typhlitis	47 (18.5%)	21 (44.7%)	26 (55.3%)	4.1 (2.1–7.9)	<0.001
**Carbapenem resistance, MIC > 8 μg/mL**	140 (55%)	83 (59.3%)	57 (40.7%)	3.9 (2.1–7.2)	<0.001
**Adequate initial empirical treatment**				1.7 (1.0–3.0)	0.05
- Yes	114 (44.8%)	87 (76%)	27 (24%)		
- No	140 (55.2%)	91 (65%)	49 (35%)		
**Time to start active treatment**				1.9 (1.1–3.4)	0.014
- Less than 48 h	147 (57.8%)	111 (75.5%)	36 (24.5%)		
- More than 48 h	107 (42.2%)	67 (62.6%)	40 (37.5%)		
**Number of active antibiotic target**				1 (0.5–1.9)	0.85
- Monotherapy	73	52 (71%)	21 (29%)		
- Combination	156	112 (73%)	42 (27%)		

**CRE**: carbapenem-resistant Enterobacteriaceae, **BSI:** Blood stream infection, **OR:** Odds ratio, **ICU:** Intensive Care Unit.

**Table 3 antibiotics-12-00405-t003:** Multivariate analysis for predictors of mortality among pediatric cancer patients with CRE-BSI.

Predictors of Mortality	*p*-Value	OR	95% CI for OR	
			Lower	Upper
- Refractory/relapse of primary malignancy	0.116	1.677	0.880	3.196
- Source (bacteremia or bacteremia with associated documented site infection)	0.758	1.159	0.453	2.963
- Time to start active treatment > 48 h	0.623	1.226	0.545	2.760
- Adequate initial empirical antibiotic treatment	0.616	1.227	0.552	2.729
- Number of active antibiotic target (combined vs. monotherapy)	0.268	0.547	0.188	1.589
- *K. pneumoniae* organism	0.020	2.180	1.133	4.196
- Associated typhlitis	<0.001	5.047	2.444	10.423
- Associated pneumonia	0.013	2.306	1.193	4.457
- Carbapenem resistance with MIC > 8 μg/mL	0.006	2.668	1.322	5.385

**OR:** odds ratio; **MIC:** minimum inhibitory concentration.

## Data Availability

The data supporting this study’s findings are available from the corresponding author (S.A.) upon reasonable request and with permission of the Children’s Cancer Hospital, Egypt.

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
