# Peer review of "Carbapenem-Resistant Enterobacteriaceae (CRE) among Children with Cancer: Predictors of Mortality and Treatment Outcome"

_antibiotics, 2023, doi:10.3390/antibiotics12020405_

Round 1

Reviewer 1 Report

In this paper, entitled “Carbapenem-Resistant Enterobacteriaceae (CRE) among Children with Cancer in Limited-resource Countries; Predictors of Mortality and Treatment Outcome” by Youssef Madney, it is suggested that Carbapenem-resistant Enterobacteriaceae (CRE) infections are associated with poor clinical outcome in paediatric cancers. The fact that patients undergoing chemotherapy are more vulnerable to infection is not surprising however this is, in my knowledge, the first retrospective study on this topics that identify the most common-associated infections, in areas with limited resources, and that are associated with poor clinical outcome.

Major

      In my opinion, this is very interesting retrospective analysis showing  that a proper evaluation of microbiome in cancer might be an important parameter to be consider at time of diagnosis for a more appropriate standard of care. However It will be useful to split patients between refractory(relapsed) and primary cancer and also, if possible, to show wether there is any difference in term of treatment efficiency between CRE-patients and CRE-free patients

Minor

If I can point an aspect that could be improved, I am not sure the relevant current literature has been fully covered in the introduction and discussion section.

The manuscript is sometimes difficult to read due to unstructured syntax and grammar. The same is for title could be simplified and I will avoid acronyms without explanations such as in line 40, for example “mainly KPC, NDM, VIM, and OXA-48 types….”. I will strongly suggest a professional english editing.

Author Response

Reviewer-1
In this paper, entitled “Carbapenem-Resistant Enterobacteriaceae (CRE) among Children with Cancer in Limited-resource Countries; Predictors of Mortality and Treatment Outcome” by Youssef Madney, it is suggested that Carbapenem-resistant Enterobacteriaceae (CRE) infections are associated with poor clinical outcome in paediatric cancers. The fact that patients undergoing chemotherapy are more vulnerable to infection is not surprising however this is, in my knowledge, the first retrospective study on this topics that identify the most common-associated infections, in areas with limited resources, and that are associated with poor clinical outcome.

Major

      In my opinion, this is very interesting retrospective analysis showing that a proper evaluation of microbiome in cancer might be an important parameter to be consider at time of diagnosis for a more appropriate standard of care. However It will be useful to split patients between refractory(relapsed) and primary cancer and also, if possible, to show wether there is any difference in term of treatment efficiency between CRE-patients and CRE-free patients?

  • Regarding disease status, in univariate analysis , patients with refractory (relapsed) disease, had a higher day 30 mortality ( OR:2.2 , 95% CI:1.2-3.8, p=0.005), however on multivariate logistic regression analysis; the independent factors that significantly affect the outcome of episodes (survival at 30 days) were isolation of Klebsiella (OR: 2.2, 95% CI: 1.1-4.1, p=0.020), typhlitis (OR: 5.1, 95% CI: 2.4-10.4, p<0.001), pneumonia (OR: 2.3, 95% CI: 1.2-4.5, p=0.013) and the carbapenem resistance ((OR: 2.7, 95% CI: 1.3-5.4, p=0.006).

Minor

If I can point an aspect that could be improved, I am not sure the relevant current literature has been fully covered in the introduction and discussion section.

The manuscript is sometimes difficult to read due to unstructured syntax and grammar. The same is for title could be simplified and I will avoid acronyms without explanations such as in line 40, for example “mainly KPC, NDM, VIM, and OXA-48 types….”. I will strongly suggest a professional english editing.

       a professional English editing done and explanations for acronyms were done.

Reviewer 2 Report

The manuscript presents “Carbapenem-Resistant Enterobacteriaceae (CRE) among Children with Cancer in Limited-resource Countries; Predictors of Mortality and Treatment Outcome” The author explained the prevalence of CRE in Enterobacteriaceae that will be beneficent for researchers, physicians, and others to overcome this issue and adopt an alternative way to combat these challenges. The author should give the data of antimicrobial susceptibility testing against other antibiotics except carbapenem to make this study more interesting and fruitful. 

·         I am highlighting a few mistakes as the author should writhe the name of all bacteria in italic.

·         The title should concise, accurate and informative. Please revise this.

Introduction:

·         The introduction is too general: Report the epidemiology of carbapenemases and ESBLs that focus on the study area. Please show the study gaps.

·         Line 34, 36: First, please write down the name of all bacteria in italic. Secondly, use the full name of the bacteria first i.e., Staphylococcus aureus (S. aureus), and then you can use the short name of the bacteria i.e., S. aureus. Enterobacteriaceae?

·         Line 41: Carbapenem-resistant Enterobacteriaceae (CRE) is defined prior to using an acronym. Just write “CRE”. Please revise this.

·         Line 37: Salmonella and Shigella?

·         Line 44-49: Not clear. The author should rewrite and explain it clearly.

Results:

·         The author should present the data in a simple way by using a heading. Please revise this portion accordingly.

·         Variable should present data as “For example: [E. coli 50% (6/12)] in the whole manuscript”

·         Discussion should be revised according to the result.

·         Conclusion is not coming from results and Discussion. Please revise this.

Material and Methods:

·         Section 4.1: The author should explain about disk diffusion method either used for measurement of a zone of inhibition or etc

·         Section 4.2: IDSA is not defined prior to using acronyms.

·         Not clear regarding sample collection, transportation, and preservation: How did you collect the blood sample? How did you transport the samples into the microbiology lab? Did you preserve the sample? If yes, then for how many days, and what was the temperature?

·         Line 251: Provide the manufacturer details of each chemical/material used (Company, City, Country).

·         Columbia blood agar and MacConkey?

·         Please give a reference to why the author only uses Columbia blood agar and MacConkey agar for sample inoculation.

·         35°C    OR 37°C? Please correct this.

·         The author should explain Disc diffusion and MIC used for which purpose in this study.

Author Response

Reviewer (2)

Suggestions for Authors

  • The manuscript presents “Carbapenem-Resistant Enterobacteriaceae (CRE) among Children with Cancer in Limited-resource Countries; Predictors of Mortality and Treatment Outcome” The author explained the prevalence of CRE in Enterobacteriaceaethat will be beneficent for researchers, physicians, and others to overcome this issue and adopt an alternative way to combat these challenges. The author should give the data of antimicrobial susceptibility testing against other antibiotics except carbapenem to make this study more interesting and fruitful. 
    • Carbapenem resistance was reported in all patients with MIC less than 8 μg/mL (45%) and more than 8 μg/mL (55%). Aminoglycoside resistance was reported in 46.5%, and 87% were resistant to quinolones. Colistin resistance was reported in 1% (two isolates), while no resistance was reported to tigecycline. (added to main document, results part)
  • I am highlighting a few mistakes as the author should writhe the name of all bacteria in italic.
  • The title should concise, accurate and informative. Please revise this.
    • Done and corrected

Introduction:

  • The introduction is too general: Report the epidemiology of carbapenemases and ESBLs that focus on the study area. Please show the study gaps.
    • Introduction part modified, added paragraph about epidemiology of CRE and different carbapenems genes according to different geographic distributions. also study gaps are highlighted.
    • Line 34, 36: First, please write down the name of all bacteria in italic. Secondly, use the full name of the bacteria first i.e., Staphylococcus aureus( aureus), and then you can use the short name of the bacteria i.e., S. aureusEnterobacteriaceae?
    • Corrected and added.
    • Line 41: Carbapenem-resistant Enterobacteriaceae(CRE) is defined prior to using an acronym. Just write “CRE”. Please revise this.
    • Corrected and added.
    • Line 37: Salmonella and Shigella?
    • Most identified Enterobacter sp in our study was Enterobacter cloaca.
    • Line 44-49: Not clear. The author should rewrite and explain it clearly. Rewriting the paraph done and modified

Results:

  • The author should present the data in a simple way by using a heading. Please revise this portion accordingly.
  • Done and corrected
  • Variable should present data as “For example: [ coli50% (6/12)] in the whole manuscript”
  • Done and corrected
  • Discussion should be revised according to the result.
  • Done and correct
  • Conclusion is not coming from results and Discussion. Please revise this.
  • Done and corrected

Material and Methods:

Section 4.1: The author should explain about disk diffusion method either used for measurement of a zone of inhibition or etc

Carbapenem resistance was defined according to the Clinical and Laboratory Standards Institute (CLSI) criteria. Disc diffusion not used, only MIC (highlighted in methodology)

Section 4.2: IDSA is not defined prior to using acronyms.

The Infectious Diseases Society of America (IDSA)Corrected and added to main document

Not clear regarding sample collection, transportation, and preservation: How did you collect the blood sample? How did you transport the samples into the microbiology lab? Did you preserve the sample? If yes, then for how many days, and what was the temperature? 

Line 251: Provide the manufacturer details of each chemical/material used (Company, City, Country).

Columbia blood agar and MacConkey?

Blood cultures:  Pediatric blood culture bottles (BACTECTM PedsPlusTM/F, BD Diagnostics & BacT/ALERT®PF Plus) inoculated with patients' blood were loaded in the corresponding blood culture system. After flagging, positive blood culture broths were sub-cultured on Columbia blood agar and MacConkey No.2 agar plates (Oxoid™) and incubated at 37° C. 

Please give a reference to why the author only uses Columbia blood agar and MacConkey agar for sample inoculation.

 Karen C. Carroll, Melvin P. Weinstein. Manual and Automated Systems for Detection and Identification of Microorganisms. In: Manual of Clinical Microbiology, 9th, Patrick R. Murray (Ed), ASM Press, Washington, D.C. 2007. p.192.  Reference added

 35°C    OR 37°C? Please correct this.

Corrected = 37 C

The author should explain Disc diffusion and MIC used for which purpose in this study.

Disc diffusion not used, only MIC on CLSI criteria was used for identification and definition of the resistance. Identification: The isolated colonies were identified using the MALDI-TOF Vitek MS IVD system (BioMérieux; Marcy l’Etoile, France).  

Antibiotic susceptibility testing: Automated Vitek 2 Compact (BioMérieux SA, Marcy l’Étoile, France) was used on isolates on the agar plates as recommended by the manufacturer and interpreted according to CLSI guidelines.

Carbapenem-resistant Enterobacteriaceae (CRE) definition: Any Enterobacteriaceae that test resistant to at least one of the carbapenem antibiotics with minimum inhibitory concentration (MIC) ≥ 4 μg/mL for meropenem, imipenem, or doripenem, or MIC ≥ 2 μg/mL for ertapenem.

So MIC was used for both definition of resistance and applied also for treatment choice according to MIC of carbapenem. Patients with CRE – BSI were treated according to carbapenem MIC. For MIC ≤ 8 μg/mL, high dose continuous infusion (40 mg/kg over 3 hours infusion) was the main active treatment with adding a second active agent (aminoglycosides or quinolones if sensitive or colistin if both were resistant) (Added to methodology part)

Reviewer 3 Report

Thanks for this study. CRE is becoming a significant public health concern, therefore your retrospective cohort study is definitely interesting. However, there are some flaws in the methodology and presentation in the current state, which I therefore recommend to reconsider. I outline my concerns below.

Major concerns:

- MIC<8 is quite confusing to read, since CRE is defined as either >4 or >2 if ertapenem. Therefore, please replace this by MIC 4-8. 

- For the logistic regression analyses, you should disclose details in terms of the outcome parameter, criteria to select the best model, measures of goodness of fit (e.g., AUROC, deviance statistic).

- Please outline what the practices for CRE colonization screening were during the study time. 

- It would be helpful to report the individual MIC per organism with mean, median and spread.

- Table 2: is there a reason why the OR are not reported? That would be helpful to compare with the multivariate analysis.

Minor concerns:

- Line 47-48: this sentence needs a reference. Please outline what you mean with delayed inappropriate therapy- e.g., delay of targeted therapy?

- Please review the spelling of bacteria in a correct (in Italics) and consistent way (e.g., line 65-67: "E. coli vs. klebsiella pneumonia" schould be either "Escherichia coli vs. Klebsiella pneumoniae" or "E. coli vs. K. pneumoniae"

- Table 1: the age distribution is hard to assess without the full data. Therefore, please report ages as per FDA pediatric age categories. Furthermore, please specifty the solid tumor diagnoses.

- Line 68: please specify if these statistics apply to the cohort of children suffering CRE.

Author Response

Reviewer 3

Thanks for this study. CRE is becoming a significant public health concern, therefore your retrospective cohort study is definitely interesting. However, there are some flaws in the methodology and presentation in the current state, which I therefore recommend to reconsider. I outline my concerns below.

Major concerns:

- MIC<8 is quite confusing to read, since CRE is defined as either >4 or >2 if ertapenem. Therefore, please replace this by MIC 4-8. 

- done and corrected

- For the logistic regression analyses, you should disclose details in terms of the outcome parameter, criteria to select the best model, measures of goodness of fit (e.g., AUROC, deviance statistic).

- In logistic regression model, the outcome parameter was in hospital mortality. The best fit model was selected based on Hosmer-Lemeshow test and deviance as measure of goodness of fit. The former was insignificant with a p-value of 0.401 and the latter was 137.92 compared to 197.75 for the null model.

- Please outline what the practices for CRE colonization screening were during the study time. 

- During the study period, surveillance for CRE colonization screening was not routinely done. However, in this study, 30% of patients with CRE -BSI infections were colonized with CRE. Also, a high mortality rate was reported among pediatric cancer patients with CRE (30%), especially if delayed adequate active treatment was given. Based on data reported from this study, colonization surveillance with a rectal swab for all leukemia patients admitted to our center with the use of colonization-guided empirical antibiotics was implemented.

- It would be helpful to report the individual MIC per organism with mean, median and spread.

Correlation between MIC with type of organism was reported in our study and was found to be of clinical significance. Patients with K. pneumonia were reported to have a higher MIC to carbapenem (80% versus 37%). Higher K. pneumonia MIC also was associated with poor prognostic factors like documented site infection (pneumonia and typhlitis), delayed microbiological clearance (30% versus 16% for E. coli) and more presentation with septic shock and need for ICU admission (42% versus 19%). (Added to main document)

- Table 2: is there a reason why the OR are not reported? That would be helpful to compare with the multivariate analysis.

Done and added

- In univariate analysis, we found many risk factors , so We repeated selected the most important risk factors and removed factors measuring similar things  (table 2).

  • Only Eight (8) factors were selected, 7/9 factors were statistically significant and tested on multivariate analysis.
  • ICU admission and inotropic support was considered as a result (rather than being itself as cause or predictor) and excluded from multivariate analysis to see the impact of other causal variables).

Minor concerns:

- Line 47-48: this sentence needs a reference. Please outline what you mean with delayed inappropriate therapy- e.g., delay of targeted therapy?

- corrected to delayed targeted therapy

- Please review the spelling of bacteria in a correct (in Italics) and consistent way (e.g., line 65-67: "E. coli vs. klebsiella pneumonia" schould be either "Escherichia coli vs. Klebsiella pneumoniae" or "E. coli vs. K. pneumoniae"

- done and corrected

- Table 1: the age distribution is hard to assess without the full data. Therefore, please report ages as per FDA pediatric age categories. Furthermore, please specifty the solid tumor diagnoses.

- age modified in the table according to FDA

- Line 68: please specify if these statistics apply to the cohort of children suffering CRE.

The statistics applied in this part included all gram-negative bacteremia, not only CRE.

Round 2

Reviewer 2 Report

The revised manuscript looks good. I found a few typos and errors in the manuscript. Please look at this carefully.

Line 15: Enterobacteriaceae? please write in italic.

Reviewer 3 Report

Recommend a last spelling check.